# Structural, Personal and Socioenvironmental Determinants of HIV Transmission among Transgender Women in Indonesia

**DOI:** 10.3390/ijerph18115814

**Published:** 2021-05-28

**Authors:** Nelsensius Klau Fauk, Maria Silvia Merry, Theodorus Asa Siri, Lillian Mwanri, Paul Russell Ward

**Affiliations:** 1College of Medicine and Public Health, Flinders University, GPO Box 2100, Adelaide, SA 5001, Australia; fauk0001@flinders.edu.au (N.K.F.); lillian.mwanri@flinders.edu.au (L.M.); 2Institute of Resource Governance and Social Change, Jl. R. W. Monginsidi II, No. 2, Kupang, Nusa Tenggara, Timur 85221, Indonesia; 3Medicine Faculty, Duta Wacana Christian University, Jl. Dr. Wahidin Sudirohusodo, No. 5-25, Kotabaru, Yogyakarta 55224, Indonesia; silvia.tropmed@gmail.com; 4Saint Peter Pastoral Institute of the Diocese of Atambua, Kefamenanu, Nusa Tenggara, Timur 85613, Indonesia; asasiritheo@yahoo.com

**Keywords:** HIV infection, structural personal and socioenvironmental factors, transgender women, *waria*, Indonesia

## Abstract

Transgender populations are considered as a highly vulnerable group to HIV infection. This study aimed to understand structural, personal and socioenvironmental factors and the mechanisms through which these factors facilitate HIV transmission among transgender women (*waria*) in Yogyakarta, Indonesia. A qualitative inquiry using one-on-one in-depth interviews was employed to collect data from participants (*n* = 29). Thematic analysis was used to guide data analysis. Findings showed that poverty in families, a sense of responsibility to support family necessities, limited employment options and low education attainment were the structural factors driving participants’ engagement in sex work practices and unprotected anal intercourse, which facilitated HIV transmission among them. Personal need fulfilment and the desire for savings were personal factors driving their engagement in these high-risk practices that supported HIV transmission. Social relationships, social influence and the participants’ living environment were socioenvironmental factors that also supported sex work practices and HIV transmission among the participants. The findings indicate the need for capacity building in terms of knowledge and skills for *waria* populations to prepare and enable them to gain meaningful employment to prevent the vicious cycle of HIV transmission among them. As structural factors seemed to be the main drivers predisposing *w**aria* to HIV acquisition, further studies to explore effective HIV/AIDS interventions that address economic aspects of *waria* in Yogyakarta and other similar settings in Indonesia are recommended.

## 1. Introduction

Globally, transgender populations are considered as a group at high risk for HIV infections [1,2,3,4]. Compared to the prevalence of 0.15% in the general Indonesian population, the prevalence of HIV infection among transgender populations is the highest among other countries in Asia and the Pacific region, accounting for 24.8% [5]. However, the available reports from the UNAIDS and the Indonesian Ministry of Health provide HIV data on transgender populations in general, without differentiating the data on transgender men from those of transgender women in Indonesia [4,5].

The term transgender in Indonesia has often been used to refer to, and is also known as, *waria*, a combination of two Indonesian words: *WAnita* (woman) and *pRIA* (man). *Waria* are men who adorn themselves to appear as women or male individuals who generally dress and act in a normatively feminine manner, with previous studies referring to them as “male-to-female transgenders” or transgender women [6,7,8,9,10]. *Waria* are viewed as sexual deviants, people who are contaminated, and are often rejected in most cultures and societies in Indonesia [6,7,8,9,10]. Such views are reported to lead to stigma associated with being *waria*, which prevents them from talking openly about their health status or seeking healthcare services [11,12,13]. The term transgender women and *waria* will be used interchangeably throughout the manuscript. The term transgender women will be used once discussing the international literature, and the term *waria* will be used to maintain the local meaning attached to it once reflecting the participants’ self-identity or referring to participants’ perspectives.

Engagement in unprotected anal intercourse (UAI) with multiple sex partners, in short-term relationships and in sex work, is the main behavioural factor that facilitates HIV transmission among the transgender women population across the globe [14,15,16,17]. Needle sharing for injecting drugs and female hormones is also a behavioural factor facilitating the spread of the infection among them [16,18,19]. Needle sharing for transgender women is deemed necessary due to poor access to sterile needles when in immediate need and the fear of being caught by police when buying needles [16,18,19]. A lack of knowledge about how HIV is transmitted and prevented is also another important supporting factor for their engagement in both UAI with multiple sex partners and needle sharing practices without being aware of the risk of HIV transmission [15,18,20]. Poverty and poor economic conditions are also drivers of their involvement in sex work and UAI in exchange for money, making them highly susceptible to HIV infection [16]. Additionally, having a history of sexually transmitted infections (STIs) also facilitates HIV transmission among this population [21].

Although the HIV prevalence among transgender populations in Indonesia continues to increase, evidence on factors enabling HIV transmission and the mechanisms through which these factors facilitate HIV transmission among *waria* is still limited. Previous studies have investigated HIV transmission among *waria* in Indonesia; however, they have solely focused on individual level factors, such as knowledge of HIV/AIDS, sexual behaviours and condom use [7,8,10,22]. Therefore, this study aimed to understand broader risk factors for HIV transmission by exploring structural, personal and socioenvironmental factors and the mechanisms through which these factors support HIV transmission among *waria* in Yogyakarta, Indonesia.

## 2. Methods

### 2.1. Conceptual and Theoretical Frameworks

The Social Determinants of Health (SDH) framework was used to guide the conceptualisation and discussion of the study findings [23]. The SDH framework recognises the important role that social, economic and structural factors play in increasing populations’ vulnerability to poor health outcomes. This framework suggests that structural factors, including poor economic conditions, unemployment and low education level, are determinants of poor health [23]. These factors increase populations’ vulnerability to poor health outcomes, including HIV infections, through mechanisms, such as engagement in sex work in exchange for money and UAI due to lack of power to bargain safe sex activity. The SDH framework in the current study considers the multiple factors attributable to HIV transmission in *waria* populations. In addition to the SDH framework, some constructs of the Social Networks Theory (SNT) [24] were also used to examine the interactions and relationships among *waria* in the current study setting. The SNT suggests that social networks and the socioenvironment where people live and interact play an important role in people’s social issues, including health outcomes. The theory informs how cultural perspectives and norms can shape social networks that facilitate social influence, which can have negative impacts on behaviours and health outcomes of the population groups. Cultural perspectives and norms can also contribute to shaping characteristics of network ties [24]. Social networks’ structures and characteristics could also provide opportunities for social interactions and influences among people through connecting casual sex clients, supporting engagement in casual sex and UAI and in concealing the sexual orientation, hence increasing their vulnerability to HIV transmission and acquisition [12,24]. The SNT is also applied and considered necessary to conceptualise and discuss the current study’s finding as the process of social influence explained in this theory is not explicitly explicated in the concepts of SDH framework, hence this theory helps provide a strong conceptual underpinning to understand complex social influence among *waria* as presented in the current findings.

### 2.2. Participants Recruitment Procedure

*Waria* living with HIV (*n* = 29) participated in this qualitative inquiry, which was conducted from December 2017 to February 2018 in the city of Yogyakarta, Indonesia. The participants were recruited using purposive and snowball sampling techniques. Initially, the head of a *waria* non-governmental organisation (NGO), who is also a *waria*, in the study setting was purposively approached by field researchers (NKF and MSM) to discuss about the study and the intention to recruit *waria* for interviews. Upon the discussion, she agreed to help distribute to potential participants the study information packs containing a brief about the study and the contact details of the field researchers. The information packs were distributed to potential participants through an information board at a shelter for *waria*, which belonged to the NGO. Initial participants who contacted the researchers and took part in the interviews were also requested to help distribute the information packs. This process was recursive and led to a total of 32 participants expressing interest to participate, but three did not attend interviews due to personal reasons. The recruitment ceased once the researchers felt that the information or responses of the last few participants to the interview questions were similar to those of previous participants and no new additional information was provided, an indication of data saturation. The recruitment of the participants was based on several inclusion criteria including: (I) self-identification as *waria,* (II) self-reporting as HIV positive, and (III) being 18 years old or above.

### 2.3. Data Collection and Ethical Consideration

Data were collected using one-on-one in-depth interviews by the field researchers at a researcher–participant mutually agreed time and place which was in participants’ houses or a private room at the shelter for *waria*. Interviews took between 45 and 90 min and were audio recorded digitally, and notes were taken during the interviews. The interview questions were focused on several key areas, including employment options available for *waria*; the participants’ involvement in sex work and the environment under which they do the work; economic condition of the participants and their families and its influence on their work; education background of the participants and its influence on their access to jobs and information; availability of HIV-related information and health services; the social relationships and influence among *waria*; social interaction of the participants with the members community where they lived. Although participants were offered the opportunity to read their transcripts, none of them took up this offer. Interviews were conducted in Bahasa, the national language of Indonesia, which is the primary language of the researchers and widely spoken in the study setting. Only a researcher and participant were present in the interview room and there was no established relationship between the researchers and any of the participants prior to the study. No repeated interviews were conducted with any of the participants. Each participant was provided with IDR 75,000 (± USD 5) to reimburse them for their time and transport.

Prior to the interviews, participants were advised that ethics approval for this study was obtained from the Health Research Ethics Committee, Duta Wacana Christian University, Indonesia (ref: 558/C.16/FK/2017). The procedures used in this study adhered to the tenets of the Declaration of Helsinki. The participants were also informed about the purpose of the study and that their participation in this study was voluntary. They were advised that they had the right to withdraw from participating during the interview without any consequences should they have felt uncomfortable with the questions/topics being asked about. They were assured that their personal identity would be made confidential and anonymous by assigning a unique Study Identification Number (P1, P2, …) to prevent the possibility of linking back the data or information to any individual in the future. Each participant signed a consent form and returned it to the researcher on the interview day.

### 2.4. Data Analysis

After the verbatim transcription of the recorded interviews by the field researchers (NKF, MSM), data were thematically analysed using the steps introduced by Braun and Clarke [25]. To maintain the quality and validity of the data, cross check and comparison of the transcription and translation between the two authors were performed during the transcription and translation process. Data were further checked for clarity of transcription and accuracy of translation by other authors. Although, data analysis was primarily undertaken by NKF and MSM, team-based discussion of the findings was undertaken, and team decisions were made about the validity of the final themes and interpretation. Braun and Clarke’s thematic analysis proposes six steps to analyse qualitative data: (I) familiarisation with the data through reading the transcripts repeatedly to search for meanings, patterns and ideas; (II) generation of initial codes followed by close coding through which codes to data extracts from each data item were connected or compared; (III) searching for candidate themes and sub-themes by sorting different codes into potential themes and sub-themes; (IV) reviewing and refining the candidate themes and sub-themes by reading all the collated extracts for each theme and sub-theme to see whether or not they appeared to form a coherent pattern; (V) defining and naming the themes and sub-themes through the identification of the essences of what each theme was about (as well as themes overall) and the determination of what aspect of the data each theme captured; and (VI) producing the report.

## 3. Results

### 3.1. Participants’ Sociodemographic Information

The mean age of the study participants was 44 years. The participants were from eight different provinces, with the majority coming from the Special Region of Yogyakarta (38%), Central Java (21%), and West Java (14%) provinces. A small proportion of participants were from the North Sumatera (7%), South Sumatera (7%), East Java (7%), Riau Islands (3%), and Bengkulu (3%) provinces. All the participants had been involved in sex work for years and diagnosed with HIV. Several participants self-reported to have had one or two STIs including syphilis, gonorrhoea, and genital warts. A few of them self-reported to have been diagnosed with tuberculosis. All the participants had been on antiretroviral therapy (ART) for several years. The majority of them were high school graduates (62%), followed by elementary school graduates (21%), and the rest did not complete elementary school (17%).

### 3.2. Structural Factors

#### 3.2.1. Poor Economic Condition of *Waria’s* Families

Poverty in families played a crucial role as a driving factor for participants’ engagement in sex work practices. A few participants reported having engaged in commercial sex work, albeit at an early age due to poor economic conditions within their families, a practice that transmitted HIV infection among them. Such “choice” seemed to be made based on several reasons, including the desire to be independent and the awareness that they would not further their education to the higher level due to the poor family economic condition:

*“Our family life [economic condition] was very difficult in the village, that was the reason I decided to take care of myself and the work I could do was to be a ‘kupu-kupu malam’ [sex worker, literally means night butterfly]”*.(P29, 36 years old, junior high school graduate)

*“After graduating from junior high school, I started going out at night [being a sex worker] because I was aware that the economic condition of my family was difficult, and my parents could not afford to support my further schooling. I wanted to be independent and it [sex work] was the easiest work [available job she could find] I could do at that time...”*.(P14, 41 years old, junior high school graduate)

The willingness to assist the family members and to alleviate the economic burden of their parents was also a significant factor that necessitated *waria* to engage in sex work practices. Several participants described that responsibility for family and the need to help family members or parents in a range of situations such as medical and education costs, and other needs underpinned their decision to sell sex for money, another mechanism through which poor family economic condition contributed to HIV transmission and acquisition among them:

*“I work like this [selling sex] from afternoon to dawn because I need to help the necessities of my family and to alleviate my parents’ economic burden. You know the whole night is the time for me to work as a ‘kupu-kupu malam’”*.(P2, 51 years old, did not complete elementary school)

*“One of the reasons I decided to be a sex worker is to help my family, my dad and mom got sick very often and couldn’t work. They passed away few years ago. I do this because I want help my parents and my brothers and sisters”*.(P23, 42 years old, senior high school graduate)

#### 3.2.2. Limited Employment Opportunities

The lack of employment opportunities for *waria* was indicated as a contributing factor for their engagement in sex work and the spread of HIV infection among them. The participants’ decision to engage in sex work as a livelihood option due to the lack of employment opportunities available for them and unacceptance of their *waria* status was the mechanism through which the limited employment opportunities contributed to or facilitated the further spread and acquisition of HIV among them and their sex clients:

*“Many waria choose to become sex workers because this is the easiest job we can do even though it makes us vulnerable to HIV. We do not have many choices like others [non-waria]. I do not see any opportunities for us in terms of job”*.(P23, 42 years old, senior high school graduate)

*“I have engaged in this work [sex work] for many years because I do not find other suitable jobs. …. Many institutions do not accept waria”*.(P1, 55 years old, did not complete elementary school)

In addition, it was also indicated that the participants’ low level of education attainment was one of the limitations in securing other jobs available in the study setting. The participants acknowledged that low level of education had led to ineligibility to access better employment:

*“I do not have many [employment] options to choose. I realise that because I do not have higher education background, I cannot access better employments”*.(P18, 33 years old, elementary school graduate)

*“It is very difficult for me to find a job, I am not eligible to get a good job because I do not have diploma or undergraduate certificate”*.(P6, 50 years old, junior high school graduate)

#### 3.2.3. Low Education Attainment Level

A low level of education which led to poor health literacy was also indicated as a factor that contributed to the participants’ lack of knowledge about the means of HIV transmission and prevention. All the participants described that they had a low level of education, which seemed to be a reason why they were not aware of the HIV issue and condoms prior to the HIV diagnosis:

*“I did not even complete elementary school and you know the level of knowledge of people who do not attend formal education like me and who live in the villages. I did not know anything about HIV before I was tested, and I did not search for information about it [HIV] because I did not know anything about it back in my village. I did not even know how and where to search information. So, the thoughts about HIV or condoms did not come to my mind even though I engaged in unprotected sex for many years with different people”*.(P2, 51 years old, who did not complete elementary school)

*“When I was at junior high school, topics like HIV or condoms were not taught, so I did not know anything at all about them. If I continued my study to university level, then I might have known and learned about these things since long time ago. I gained information about HIV, condoms and ARV therapy after the HIV diagnosis”*.(P13, 47 years old, junior high school graduate)

Unavailability of HIV-related information within communities where the participants lived was also a supporting factor of the spread of HIV infection among them. It was indicated as a condition that led to the lack of knowledge of HIV and unawareness of participants about the possibility of HIV transmission and acquisition through sex work practices or UAI. The participants described that they knew nothing about HIV before they moved to Yogyakarta and underwent HIV test due to unavailability of information related to HIV in the villages where they previously lived:

*“There was no information about HIV/AIDS or condoms at all. So, I did not know anything about HIV or condoms back in those days [before she moved to Yogyakarta]. There was not any HIV programs or activities. Now, nurses, doctors and NGOs create HIV programs and carry out activities to help HIV positive people or to educate people within communities. So, at the beginning of my engagement in this work [sex work], I was not scared of HIV and did not use condoms because I was not aware of these things. I first heard about HIV/AIDS and condoms from my friends [waria] once I moved here [Yogyakarta] and was tested HIV positive. Now, most of the time I do not use [condoms] once having sex with clients because they do not want to use, but what I can do is adhere to ARV therapy so that my viral load is undetected [suppressed] and I do not transmit it to them [sex clients]”*.(P16, 55 years old, junior high school graduate)

### 3.3. Personal Factors: Personal Needs and Savings for the Future

The participants’ personal needs and motivation to have savings were reported to have a significant influence in supporting their engagement in sex work practices. For example, the need to support everyday life or to afford to buy food was described as one of the main reasons they engaged in sex work which was seen as a quicker way to earn money:

*“I need cash to fulfil my essential needs, so I do this [selling sex] because this the quicker way for me to earn money to support my life everyday”*.(P15, 33 years old, junior high school graduate)

Savings for the future was another instance of motivating factors for the participants’ engagement in sex work. The stories of a few participants showed that they kept engaging in sex work practices due to being motivated to earn more and make some savings for the future. The following quote illustrated this assertion:

*“…. I think of my future as well, that is why I am motivated to continue engaging in this work [sex work]. I want to earn more and save some amount I earn every night for my future. I am aware that this is a very risky work and I will not be strong forever to do this. I save some mount so that I can use to support my life once I am old and stop doing this”*.(P17, 44 years old, elementary school graduate)

Participants’ dire need to earn and support their need and the need of others significantly compromised their bargaining power for safe sex practices. These factors also rendered the inability of participants to negotiate for high rates with their sex clients:

*“If I propose it [condom use] they [sex clients] would not agree because they spend money, they pay, so they do not want to use, and I cannot do anything. On the other side, I also need to earn to support my life and help the others [her families]. So, I cannot put higher price either. … I may have got it [HIV] from any of them because of that [lack of condom use]”*.(P5, 39 years old, senior high school graduate)

*“I am trying to earn money every day so every time I have a client, I always ask how much money he has, if he has IDR 20,000 or IDR 30,000 (±USD 1.5 or 2) then I am okay with that. If I put the price a little bit higher and he can’t afford it then I will not get anything, I need money to support my life”*.(P2, 51 years old, who did not complete elementary school)

### 3.4. Socioenvironmental Factors

#### 3.4.1. Social Relationships and Influence

The participants in this study reported that *waria* had good social relationships among them, reflected in the fact that they knew each other “*we [waria] know each other very well because we meet almost every day,*” (P25, 42 years old, junior high school graduate) and were aware of the sex work practices each of them engaged in “*all my friends ‘go out every night’ [selling sex]*” (P17, 44 years old, elementary school graduate). The social relationships among them and the knowledge about the sex work practices *waria* populations engaged in seemed to be used by the participants to reinforce their engagement in sex work which facilitated HIV transmission among them:

*“All waria in the city are well connected to each other …. We meet each other almost every night at the meeting points [places to meet sex clients] …. It seems like all of us [waria] do this [engage in sex work], it is common among us. I am very sure all of us [waria] get HIV because of our work”*.(P4, 39 years old, senior high school graduate)

*“To be honest, all waria here go out every night [selling sex] to meet clients. Everyone [waria] is doing this [sex work], so I just do it, just like the others. Now I am still doing it [selling sex] even though I have been diagnosed with HIV. I may have got it from any of the clients, but I do not know who he is”*.(P10, 53 years old, senior high school graduate)

Social relationships and interactions among *waria* were also reported to facilitate social influence among them in terms of introduction to and engagement in sex work, a mechanism through which social relationships and interactions contributed to the spread of HIV among them. This was reflected in the following quotes:

*“Initially I was asked by a friend of mine [waria] to accompany her to the meeting points with her [sex] clients. So, I know about this [sex work] through her. I accompanied her a few times and then I started to do this [selling sex]. Finally, I get this HIV”*.(P27, 39 years old, junior high school graduate)

*“I was told by a friend mine [waria] about this [sex work], she told me how much she earned every night, so I was tempted by the money and got involved in it. …. This [sex work] made me get HIV because clients do not want to use condoms”*.(P3, 41 years old, who did not complete elementary school)

Furthermore, frequent interaction with their sex clients was reported to facilitate social relationships between *waria* who participated in this study and their sex clients. All the participants described that they and their sex clients knew each other due to the frequent meeting at the meeting points. This was described to lead to the exchange of personal contact details that facilitated meeting and sexual transaction between them and their sex clients:

*“As I have been involved in this kind of work [sex work] for many years, I am familiar with many faces at the meeting spots. I often exchange mobile number with them [sex clients], so they often call me once they need me [to have sex]. It helps us easily reach each other”*.(P12, 50 years old, 50 years old, senior high school graduate)

*“I go out every night [to meet clients at the meeting points] and have sex with different persons every night. So, I know many of them [sex clients], some clients exchange contact numbers with me and they can contact me at any time [for sex purposes)”*.(P14, 41 years old, junior high school graduate)

#### 3.4.2. The Living Environmental of the Participants

The environment or the surroundings where the participants lived and interacted also played an important role in their engagement in sex work practices and UAI through which they acquired HIV infection. The availability of meeting points with sex clients and hotel rooms for short-term rent for sex with clients were some instances of environmental factors that facilitated sex work practices among *waria*, another mechanism showing how environmental factors played a role in HIV transmission and acquisition among them:

*“There are spots to meet clients and some hotels where we can rent a room for short time [to have sex]. Everybody knows these spots and hotels”*.(P29, 36 years old, junior high school graduate)

*“There are many well-known spots here [Yogyakarta] to find clients. I just need to go there and wait for them [sex clients]. There are many of them who come to the meeting spots to look for us [waria]. …. If I get a [sex] client then we just need to go to nearby hotel to rent a room for short term [just for sex] and after that I go back to the meeting spot. People in the hotels already know if we [waria] want to rent a room”*.(P20, 37 years old, elementary school graduate)

## 4. Discussion

As recognised by the World Health Organization [26,27], transgender populations are a community that is at high-risk to HIV infection due to their engagement in high-risk behaviours such as UAI and injecting drug use (IDU). This study explores structural, personal and socioenvironmental factors that support HIV transmission and the mechanisms through which these factors contribute to HIV transmission among *waria* in Yogyakarta, Indonesia.

The current findings suggest that structural factors such as poverty in the participants’ families was a strong driver for their engagement in sex work practices and UAI, which predisposed them to HIV transmission. The findings support the constructs of the SDH framework and prior studies with various population groups [15,16,23,28], which have reported the influence of social and economic factors on people’s decision making and behaviours, such as engagement in UAI with multiple sex partners. This study also adds further evidence to the existing literature [15,16,29] by reporting *waria’s* responsibility for family needs as a driver for their engagement in sex work practices through which they acquired HIV infection. In the current study setting and as was described by the participants, the limited employment opportunity the participants had due to the low level of education and the lack of acceptance towards them in a workplace setting was another structural factor which seemed to force their “choice” or “decision” to engage in sex work practices. The findings support the reports of a previous study [30], suggesting low level of education and lack of skills for well-paid work as driving forces for women’s engagement in sex work, a high-risk supporting factor for STIs transmission among them.

A low level of education attainment, which seemed to lead to poor health literacy, including the lack of knowledge about HIV/AIDS and preventive functions of condoms, was also a structural factor that played a role in the spread of HIV among the participants, which is consistent with previous findings of studies with different population groups [30,31,32,33]. It is plausible to consider health literacy among *waria* and transgender populations in general as an important factor to address because it affects their ability to access, understand and use health information and health system successfully to manage their own health [34]. The little evidence about health literacy from across countries, ability of individuals to gain access to, understand and use information in ways in which to promote and maintain good health, suggests the need for more research to increase people’s knowledge base about health issues [35], including HIV in transgender women. The current findings also suggest that participants’ dire needs for cash to fulfil their personal needs or basic necessities and save for the future were personal factors that drove their participation in sex work, made them unable to negotiate for high rates with their sex clients and compromised their bargaining power for condom use, which placed them and their clients at high risk for HIV transmission and acquisition. These are in line with the findings of previous studies in Indonesia [29,30], reporting the choice or preference of female sex workers (FSWs) for unsafe sex with their sex clients for more economic benefits over the risk of contracting STIs.

The current findings suggest that social relationships among *waria* constitute an enabling factor for their engagement in sex work, which facilitated HIV transmission among them. Connection and interaction with each other, being aware of sex work practices and introducing sex work practices to each other were the mechanisms through which social relationships and interaction facilitated risky sexual practices or UAI that supported HIV transmission and acquisition among the participants. Given that *waria* populations are socially and culturally rejected within communities in many parts of Indonesia, it is highly likely that they build strong social networks among them to survive, share, support each other, and which also facilitate negative social influence among them, as reported by the current participants. The findings enrich the existing knowledge about factors enabling HIV transmission among transgender women, which are mainly individual-level factors, as reported in previous studies in Indonesia and globally [7,8,10,12,14,15,17,22]. The findings also confirm the constructs of the SDH framework [23] and SNT [24], which suggest that social factors, social networks and connectivity play a role in influencing people’s health outcomes through mechanisms such as risky sexual behaviours, as illustrated in the narratives of the participants in the current study. As the social relationships and encounters between *waria* and their sex clients were short-lived and only for sexual purposes, these factors were also potentially enablers for high transmission of HIV among them and their many clients. The impacts of social relationships and social influence on risky sexual behaviours that enable HIV transmission have also been reported in previous studies with other population groups in Indonesia and other settings [12,36,37,38]. However, the current findings seem to be contrary to the findings of a previous study involving newcomer FSWs in Indonesia [30], which reported a lack of social network and influence among them due to sex work-related self-stigma and discouragement from senior FSWs due to the competitive working environment they faced. The current study also reveals findings which have not been previously reported elsewhere [7,8,10,12,14,15,17,22], which suggest the availability of meeting points for sex work practices between the participants and their sex clients as part of mechanisms facilitating HIV spread among them.

*Waria* populations in Indonesia are a group that is not legally and politically recognised by the government, and their sexual orientation or homosexuality is categorised as deviant and immoral behaviour, prostitution and pornography, as stipulated in some government regulations at the national and local levels [39,40]. As consequences, these populations and their rights and needs, including health needs, are not specifically addressed in policies and programs in many parts of the country [39,40]. Given that transgender women populations in Indonesia are one of the marginalised and small communities that are socially and culturally unwelcomed in many parts of the country due to their sexual orientation [6,8,9,10], the current study findings are useful for the government of Indonesia both at the national and district levels to acknowledge and accommodate these populations at a policy level and address their specific needs through programs and interventions. This may contribute to the acceptance of *waria* populations within societies in Indonesia and the improvement of their access to employment, social and health services. Specifically, the study findings indicate the need for capacity building in terms of knowledge and skills for *waria* to prepare and enable them gain meaningful employment.

### Study Limitations

Participants of this study were recruited from the city of Yogyakarta and nearly half of them lived together in a shelter for *waria*. It is therefore possible that the study could have been under-sampled by potentially not including *waria* who were outside of the social networks of the current participants. Similarly, the use of the snowball-sampling technique for participants’ recruitment could bring about a bias where participants would have provided information about the study to only those within their networks, precluding others who were not known to them. This might have led to an incomplete overview of factors that support HIV transmission among *waria*. Therefore, the findings of the current study reflect the specific conditions and situations of *waria* in Yogyakarta, which are less likely to be generalised to *waria* in other settings in Indonesia or transgender women in different settings with different characteristics. As is the case for many qualitative studies, the current study’s findings are not meant to be generalised to all transgender women, but useful to inform the development of evidence-based policies and practices to address the needs of *waria* populations in Yogyakarta and other similar settings in Indonesia.

## 5. Conclusions

This study reports several factors that supported HIV transmission among *waria* in Yogyakarta, Indonesia. These include structural factors such as poverty, responsibility to support families, limited employment opportunities and low education attainment level, which were drivers of their participation in commercial sex work. Personal need fulfillment, savings for the future and socioenvironmental factors such as social relationships, social influence and living environment were also supporting factors for their engagement in sex work practices and HIV transmission among them. As structural factors seemed to be the main drivers predisposing *w**aria* to HIV acquisition, further studies to explore effective HIV/AIDS interventions that address economic aspects of *waria* in Yogyakarta and other similar settings in Indonesia are recommended. Similarly, future studies involving large transgender populations and exploring their needs and expectations related to socioeconomic, education, information, and health aspects are recommended as these may better inform the development of interventions targeting these populations.

## Data Availability

The data presented in this study are available on request from the corresponding author. The data are not publicly available due to restrictions set by the human research ethics committee.

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
