# Peer review of "Structural, Personal and Socioenvironmental Determinants of HIV Transmission among Transgender Women in Indonesia"

_ijerph, 2021, doi:10.3390/ijerph18115814_

Round 1
Reviewer 1 Report
This manuscript, “Structural, personal and socio-environmental determinants of HIV transmission among transgender women in Indonesia”, is of interest to health care providers, healthcare organizations and countries around the world. The strength of this manuscript is that it frames the social-environmental determinants within the context of the transgendered women who are experts of their own lived experience.
The greatest gap in this manuscript was that the research questions were not stated; and, that although most of the elements of qualitative rigour were evident, others were not.
- The authors indicated that the study aimed to contribute to existing knowledge which felt like the participants were tokens in the research endeavour. Thus, I wonder if this could be reframed so that would be some benefits of engaging in this research for the participants which is an ethical consideration.
- The one-to-one interviews were undertaken with the data being collected via the use audio-recording and notes were taken during the interview. The participants chose not to review their transcripts which is fine but is often an indicator of lack of trust by community members with the research team. I was not able to ascertain whether or not member checking with regard to the interpretation of the findings was undertaken.
- This work would have benefitted from having a transgender woman on the research team as the questions may have evolved and the trust build over time in the community would have assisted with the various aspects of the research processes.
I think that this work is very valuable for the Waria and highlights gaps in the various systems (health, education, etc.) impacted by social determinants of health; however, it also demonstrates the strong support in the community but that being said that may have changed if transgender women from outside the shelter had been interviewed. These findings also provide the Government of Indonesia to re-think how they are engaging with transgender women and provide additional supports for them so that there may be other job/work opportunities made available.
The Conclusion is usually 2-3 sentences and the other information found in the Conclusion is usually moved to the Discussion.
There are a number of grammatical/spelling errors that need to be addressed as the sentences in some cases are not complete e.g. require a preposition, adverb, etc.
Author Response
Reviewer 1:
This manuscript, “Structural, personal and socio-environmental determinants of HIV transmission among transgender women in Indonesia”, is of interest to health care providers, healthcare organizations and countries around the world. The strength of this manuscript is that it frames the social-environmental determinants within the context of the transgendered women who are experts of their own lived experience.
The greatest gap in this manuscript was that the research questions were not stated; and, that although most of the elements of qualitative rigour were evident, others were not.
Response:
- The interview questions were focused on several key areas, including employment options available for waria; the participants’ involvement in sex work and the environment under which they do the work; economic condition of the participants and their families and its influence on their work; education background of the participants and its influence of their access to jobs and information; and availability of HIV-related information and health services; the social relationships and influence among waria; social interaction of the participants with the members community where they lived.
- The authors indicated that the study aimed to contribute to existing knowledge which felt like the participants were tokens in the research endeavour. Thus, I wonder if this could be reframed so that would be some benefits of engaging in this research for the participants which is an ethical consideration.
Response:
- Although the HIV prevalence among transgender populations in Indonesia continues to increase, evidence on factors enabling HIV transmission and the mechanisms through which these factors facilitate HIV transmission among waria is still limited. Previous studies have investigated HIV transmission among waria in Indonesia, however, they have solely focused on individual level factors, such as knowledge of HIV/AIDS, sexual behaviours and condom use [7, 22-24]. Therefore, this study aimed to understand broader risk factors for HIV transmission by exploring structural, personal and socio-environmental factors and the mechanisms through which these factors support HIV transmission among waria in Yogyakarta, Indonesia.
- The one-to-one interviews were undertaken with the data being collected via the use audio-recording and notes were taken during the interview. The participants chose not to review their transcripts which is fine but is often an indicator of lack of trust by community members with the research team. I was not able to ascertain whether or not member checking with regard to the interpretation of the findings was undertaken.
Response:
- After the verbatim transcription of the recorded interviews by the field researchers (NKF, MSM), data were thematically analysed using the steps introduced by Braun and Clarke [27]. To maintain the quality and validity of the data, cross check and comparison of the transcription and translation between the two authors were performed during the transcription and translation process. Data were further checked for clarity of transcription and accuracy of translation by other authors. Although, data analysis was primarily undertaken by NKF and MSM, team-based discussion of the findings was undertaken, and team decisions were made about the validity of the final themes and interpretation.
- This work would have benefitted from having a transgender woman on the research team as the questions may have evolved and the trust build over time in the community would have assisted with the various aspects of the research processes.
Response:
- The participants were recruited using purposive and snowball sampling techniques. Initially, the head of a waria non-governmental organisation (NGO), who is also a waria, in the study setting was purposively approached by field researchers (NKF and MSM) to discuss about the study and the intention to recruit waria for interviews. Upon the discussion she agreed to help distribute to potential participants the study information packs containing a brief about the study and the contact details of the field researchers. The information packs were distributed to potential participants through information board at a shelter for waria, which belonged to the NGO.
I think that this work is very valuable for the Waria and highlights gaps in the various systems (health, education, etc.) impacted by social determinants of health; however, it also demonstrates the strong support in the community but that being said that may have changed if transgender women from outside the shelter had been interviewed. These findings also provide the Government of Indonesia to re-think how they are engaging with transgender women and provide additional supports for them so that there may be other job/work opportunities made available.
Response:
- Thank you very much for the comments.
4.1. Study limitations
Participants of this study were recruited from the city of Yogyakarta and nearly a half of the of them lived together in a shelter for waria. It is therefore possible that the study could have been under sampled by potentially not including waria who were outside of the social network of the current participants. Similarly, the use of snowball sampling technique for participants recruitment could bring about a bias where participants would have provided information about the study to only those within their networks precluding others who were not known to them. This might have led to incomplete overview of factors that support HIV transmission among waria. Therefore, the findings of the current study reflect specific condition and situation of waria in Yogyakarta, which are less likely to be generalised to waria in other settings in Indonesia or transgender women in different settings with different characteristics. As is the case for many qualitative studies, the current study’s findings are not meant to be generalised to all transgender women, but useful to inform the development of evidence-based policies and practices to address the needs of waria populations in Yogyakarta and other similar settings in Indonesia.
The Conclusion is usually 2-3 sentences and the other information found in the Conclusion is usually moved to the Discussion.
Response:
- Some information in the Conclusion has been moved to the Discussion.
Given that transgender women populations in Indonesia are one of the marginalised and small communities that are socially and culturally unwelcomed in many parts of the country due to their sexual orientation [6, 9, 23, 24], the current study findings are useful for the government of Indonesia both at national and district levels to acknowledge and accommodate these populations at policy level and address their specific needs through programs and interventions. This may contribute to the acceptance of waria populations within societies in Indonesia and the improvement of their access to employment, social and health services. Specifically, the study findings indicate the need for capacity building in terms of knowledge and skills for waria to prepare and enable them gain meaningful employment.
There are a number of grammatical/spelling errors that need to be addressed as the sentences in some cases are not complete e.g. require a preposition, adverb, etc.
Response:
- The language has been improved throughout the manuscript.
Reviewer 2 Report
Thanks for sending me this manuscript. I find the manuscript to be interesting and timely. I particularly enjoyed very strong policy implications. I have some minor comments for the authors.
1) Can you provide some policy context of transgender people in the context of AIDS/HIV in Indonesia? This would help the authors provide relevant policy recommendations based on results in Discussion.
2) I am not sure why the authors decide to integrate the social network theory as part of the theoretical approach. The social determinants of health framework acknowledges the importance of social capital, which can capture the role of social network on health outcomes. Please provide some explanation on this point.
3) What is an empirical distinction between personal and structural factors? To me, 'I need to eat every day...I do not earn money from that job' seems to be structural rather than personal. Can you provide some explanation on this point?
4) Use of qualitative approach may be considered as a limitation. Can you discuss the importance of generalizability and its relation to your findings?
Author Response
Reviewer 2:
Thanks for sending me this manuscript. I find the manuscript to be interesting and timely. I particularly enjoyed very strong policy implications. I have some minor comments for the authors.
1) Can you provide some policy context of transgender people in the context of AIDS/HIV in Indonesia? This would help the authors provide relevant policy recommendations based on results in Discussion.
Response:
- Waria populations in Indonesia are a group that is not legally and politically recognised by the government, and their sexual orientation or homosexuality is categorised as deviant and amoral behaviour, prostitution and pornography as stipulated in some government regulations at national and local levels [46, 47]. As the consequences, these populations and their rights and needs, including health needs are not specifically addressed in policies and programs in many parts of the country [46, 47]. Given that transgender women populations in Indonesia are one of the marginalised and small communities that are socially and culturally unwelcomed in many parts of the country due to their sexual orientation [6, 9, 23, 24], the current study findings are useful for the government of Indonesia both at national and district levels to acknowledge and accommodate these populations at policy level and address their specific needs through programs and interventions. This may contribute to the acceptance of waria populations within societies in Indonesia and the improvement of their access to employment, social and health services. Specifically, the study findings indicate the need for capacity building in terms of knowledge and skills for waria to prepare and enable them gain meaningful employment.
2) I am not sure why the authors decide to integrate the social network theory as part of the theoretical approach. The social determinants of health framework acknowledges the importance of social capital, which can capture the role of social network on health outcomes. Please provide some explanation on this point.
Response:
- In addition to SDH framework, some constructs of the Social Networks Theory (SNT) [26] were also used to examine the interactions and relationships among waria in the current study setting. The SNT suggests that social networks and socio-environment where people live and interact play an important role in people’s social issues including health outcomes. The theory informs how cultural perspectives and norms can shape social networks that facilitate social influence which can have negative impacts on behaviours and health outcomes of the population groups. Cultural perspectives and norms can also contribute to shaping characteristics of network ties [26]. Social networks’ structures and characteristics could also provide opportunities for social interactions and influences among people through connecting casual sex clients, supporting engagement in casual sex and UAI and in concealing the sexual orientation, hence increasing their vulnerability to HIV transmission and acquisition [12, 26]. The SNT is also applied and considered necessary to conceptualise and discuss the current study’s finding as the process of social influence explained in this theory is not explicitly explicated in the concepts of SDH framework, hence this theory helps provide a strong conceptual underpinning to understand complex influence among waria as presented in the current findings.
3) What is an empirical distinction between personal and structural factors? To me, 'I need to eat every day...I do not earn money from that job' seems to be structural rather than personal. Can you provide some explanation on this point?
Response:
- Thank you for this very detailed observation. We agree with you and have taken this quote out of personal factors.
4) Use of qualitative approach may be considered as a limitation. Can you discuss the importance of generalizability and its relation to your findings?
Response:
- Participants of this study were recruited from the city of Yogyakarta and nearly a half of the of them lived together in a shelter for waria. It is therefore possible that the study could have been under sampled by potentially not including waria who were outside of the social network of the current participants. Similarly, the use of snowball sampling technique for participants recruitment could bring about a bias where participants would have provided information about the study to only those within their networks precluding others who were not known to them. This might have led to incomplete overview of factors that support HIV transmission among waria. Therefore, the findings of the current study reflect specific condition and situation of waria in Yogyakarta, which are less likely to be generalised to waria in other settings in Indonesia or transgender women in different settings with different characteristics. As is the case for many qualitative studies, the current study’s findings are not meant to be generalised to all transgender women, but useful to inform the development of evidence-based policies and practices to address the needs of waria populations in Yogyakarta and other similar settings in Indonesia.
Reviewer 3 Report
This is a clearly written paper that is well set out and that has obvious implications for public health practice. I think the methodology is slightly mis-described (as explained below), but otherwise is clear and appropriate for this type of study. There are some aspects of the analysis and discussion that I think could be clarified: in particular I think sections that highlight a link between community tolerance of waria and the spread of HIV need to be very carefully phrased.
Detailed comments:
There’s a useful explanation of the local vernacular term ‘waria’. However, after that point the term ‘transgender woman’ and ‘waria’ are both used. In some cases that’s obviously varying depending on whether you are reflecting participants’ self-identity or discussing the international literature, but in other places (e.g. 4.1) that does not seem to be the case. I think it would be useful to explain briefly how and why you are using terminology, and explicitly clarify whether they can be treated as interchangeable, especially from the participants’ perspective – would they understand ‘waria’ and ‘transgender woman’ to have exactly the same meaning?
Section 2.4. Braun and Clarke’s approach as you set it out here is typically described as thematic analysis. Framework analysis typically refers to a more structured approach, in which you categorise data more formally into a spreadsheet or table. Although there are some overlaps, if you followed Braun and Clarke’s original methodology, thematic analysis would probably be clearer to readers.
The distinction in the analysis between structural factors, personal factors and socio-environmental factors was not always particularly clear to me – I was not sure why lack of education was classed as personal, but lack of employment opportunity as structural.
There are points within this paper – in particular Section 3.4.2 and the final paragraph of section 4 of the discussion – that could be interpreted as suggesting that community tolerance of waria and sex work is facilitating the spread of HIV. I am not especially knowledgeable about the Indonesian political/social context, but it seems to me that this message is potentially open to a hostile interpretation, for example calls for repressive action against a marginalised population. I sincerely hope that this is not the authors’ intention, and the conclusion strongly suggests that this is not what the authors wish to communicate. I would suggest reviewing these sections to be sure that your meaning is clear.
Author Response
Reviewer 3:
This is a clearly written paper that is well set out and that has obvious implications for public health practice. I think the methodology is slightly mis-described (as explained below), but otherwise is clear and appropriate for this type of study. There are some aspects of the analysis and discussion that I think could be clarified: in particular I think sections that highlight a link between community tolerance of waria and the spread of HIV need to be very carefully phrased.
Detailed comments:
There’s a useful explanation of the local vernacular term ‘waria’. However, after that point the term ‘transgender woman’ and ‘waria’ are both used. In some cases, that’s obviously varying depending on whether you are reflecting participants’ self-identity or discussing the international literature, but in other places (e.g. 4.1) that does not seem to be the case. I think it would be useful to explain briefly how and why you are using terminology, and explicitly clarify whether they can be treated as interchangeable, especially from the participants’ perspective – would they understand ‘waria’ and ‘transgender woman’ to have exactly the same meaning?
Response:
- Term transgender women and waria will be used interchangeably throughout the manuscript. Term transgender women will be used once discussing the international literature, and term waria will be used to maintain the local meaning attached to it once reflecting the participants’ self-identity or referring to participants’ perspectives.
Section 2.4. Braun and Clarke’s approach as you set it out here is typically described as thematic analysis. Framework analysis typically refers to a more structured approach, in which you categorise data more formally into a spreadsheet or table. Although there are some overlaps, if you followed Braun and Clarke’s original methodology, thematic analysis would probably be clearer to readers.
Response:
- Term thematic analysis is now used in the manuscript.
The distinction in the analysis between structural factors, personal factors and socio-environmental factors was not always particularly clear to me – I was not sure why lack of education was classed as personal, but lack of employment opportunity as structural.
Response:
- Thanks for the very detailed observation, we previously thought low education attainment level fits under personal factors at it talks about health literacy or knowledge which is an individual characteristic. However, after re-reading it carefully we think it is much better to be placed under structural factors as education is a broader aspect that shapes or structures the context for HIV risk.
There are points within this paper – in particular Section 3.4.2 and the final paragraph of section 4 of the discussion – that could be interpreted as suggesting that community tolerance of waria and sex work is facilitating the spread of HIV. I am not especially knowledgeable about the Indonesian political/social context, but it seems to me that this message is potentially open to a hostile interpretation, for example calls for repressive action against a marginalised population. I sincerely hope that this is not the authors’ intention, and the conclusion strongly suggests that this is not what the authors wish to communicate. I would suggest reviewing these sections to be sure that your meaning is clear.
Response:
- Thank you very much for this very important comment. We have decided to take this part out of the results and discussion section.